# RESIZABLE NEURAL NETWORKS

## ABSTRACT

In this paper, we present a deep convolutional neural network (CNN) that performs arbitrary resize operation on intermediate feature map resolution at stage-level. Motivated by weight sharing mechanism in neural architecture search, where a super-network is trained, and sub-networks inherit the weights from the super-network, we introduce a novel CNN approach. We present *Resizable Neural Networks* (ResizableNet), which resize feature map with arbitrary scaling factor at stage level. The inference cost of ResizableNet is equivalent to a network with a single scale because the weights of a convolutional layer are shared across diversified scales. Additionally, we provide essential techniques to build and train our propose ResizableNet. We demonstrate the effectiveness of ResizableNet on large-scale ImageNet classification with various modern network architectures such as MobileNet, ShuffleNet, and ResNet.

To go even further, we present three applications of resizable networks: 1) Resizable Augmentation (*Resizable-Aug*). We present a data augmentation technique based on ResizableNet. It obtains superior performance on ImageNet classification, outperform AutoAugment by **1.2%** with ResNet-50. 2) Resizable Network for Neural Architecture Search (*Resizable-NAS*). We introduce ResizableNet as a single network that can deploy on many different hardware to meet various efficiency constraints. 3) Adaptive Resizable Neural Network (*Resizable-Adapt*). We introduce the adaptive resizable networks as dynamic networks, which further improve the performance with less testing cost via data-dependent inference.

## 1 INTRODUCTION

Scale is a fundamental component of the physical world, and it has been an active research topic in the field of computer vision. The latest design of scales for the Convolutional Neural Networks (CNN) fall into two categories: single scale networks and multi-scale networks. In single scale networks, such as ResNet (He et al. (2016)), only a fixed scale representation is learned from the feature maps. Alternatively, in the multi-scale networks, e.g., image pyramid (Sermanet et al., 2013; Dalal & Triggs, 2005) /feature pyramid (Ghiasi et al., 2019; Chen et al., 2017a) /filter pyramid structure (Szegedy et al., 2015), features from different resolutions are fused in a network; thus the models learn various type of scales from the objects. As a result, these architectures obtain state-of-the-art performance on scale critical tasks, such as object detection (Lin et al. (2017)) and semantic segmentation (Chen et al. (2017a)). It can be observed that both single scale networks and multi-scale networks essentially perform representation learning on a predefined feature map scales. This natural and intuitive choice, despite their success on various computer vision tasks, limits the CNNs to a constrained predefined scaling mechanism for dealing with scale variation problem (He et al., 2019b).

Our approach is motivated by the recent emergence of weight sharing approach on neural architecture search (NAS) (Pham et al. (2018); Bender et al. (2018); Guo et al. (2019)), where an over-parameterized super-network is trained, and sub-networks inherit the weights from the super-network. Generally, there are a large number of architectural configurations within a super-network.

In this work, we propose *Resizable Neural Networks* (ResizableNet), which can be viewed as a super-network from a scale perspective. A ResizableNet resizes a feature map with an arbitrary scaling factor at stage level. With a sequence of scaling factors at different stages, we obtain a scaling configuration, which can be used to define the feature map resolution of a network. In other words, a ResizableNet has multiple scaling configurations. It is a generalization of modern

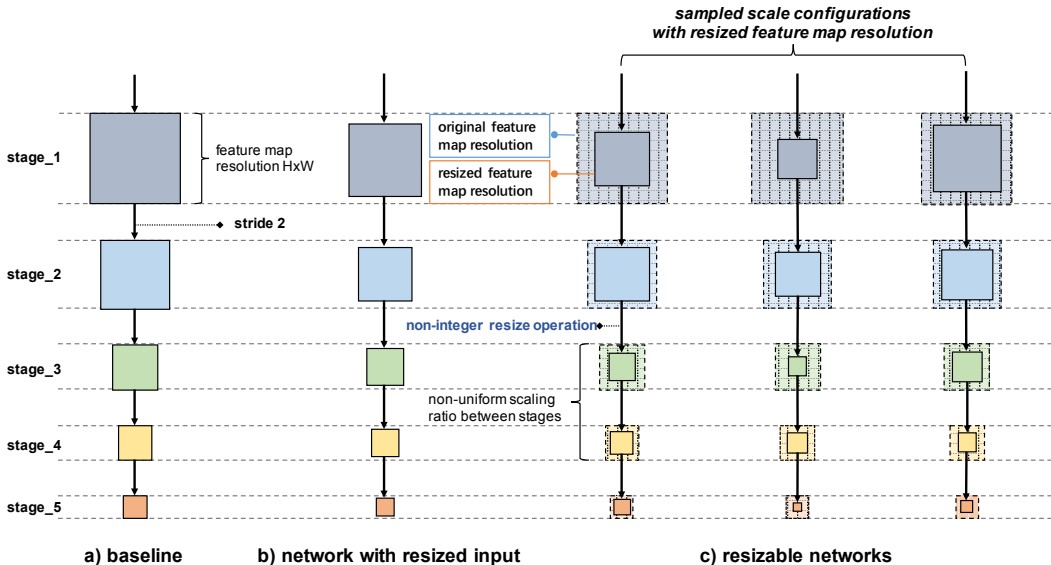

Figure 1: **Resolution scaling**. (a) is a baseline network example. (b) are conventional scaling on input resolution. (c) is our proposed resizable networks. It resizes the feature map resolution by a non-interger scaling ratio at the stage-level. A single network is trained with different scaling configurations with shared convolution. The imaginary line represents the original feature map resolution, and the full line represents the actual feature map resolution after resize operation.

CNN architecture in terms of scale, where only one fixed scaling configuration is adopted. The major advantage of ResizableNet is that the weights of a convolution layer with diversified scale factors are shared, thus substantially reduce the computational costs and memory footprints during inference. Figure 1 gives an illustration of the resizable net compared with the baseline network and input resize only network. Note that the network with resized input resolution is a special form of spatial sub-networks.

Training the resizable neural network is a non-trivial task because of two characteristics of ResizableNet. First, it requires to optimize a network with diversified feature map scales. It is challenging since there are many scaling configurations, which means it is computationally impractical to enumerate all of them, while random sampling cannot guarantee the performance of the network. Secondly, the interference in a layer between different scales leads to accuracy drops because the weights of a layer with various scales are shared in ResizableNets. We conjecture that the accuracy drop is due to the discrepancy of feature mean and variance across different scales, which leads to inaccurate batch normalization statistics. To resolve these two problems, we introduce fair sampling and scale-aware batch normalization for training the resizable network. The fair sampling samples the scaling factors multiple times of a layer in an update step without reusing the same scaling factor, thus maintain the performance of the network with different scaling configuration to some extent. The scale-aware batch normalization introduces to use a group of BN layers to replace the shared BN layer, and each scaling factor obtains an independent BN layer. Thus it solves the inconsistent BN statistics issue.

In this paper, we first validate the effectiveness of resizable neural networks compared with stand-alone networks on ImageNet. We demonstrate superior performances of Resizable Neural Network on various network architectures, such as ResNet-50, ShuffleNetV2 and MobileNetV1.

**Applications of Resizable Neural Network.** We further demonstrate three applications of ResizableNet: ResizableNet as architecture search (*Resizable-NAS*), ResizableNet for data augmentation (*Resizable-Aug*) and ResizableNet for data-dependent inference (*Resizable-Adapt*).

***Resizable-Aug.*** We introduce *Resizable Augmentation* (Resizable-Aug), a data augmentation technique based on ResizableNet. We are able to improve ResNet-50 by **2.4%** compare with baseline, which is higher than AutoAugment (Cubuk et al. (2018)). Additionally, combine our augmentation technique on a new baseline model, we obtain a model achieve **77.8%** with **365M** FLOPs. This is significantly better than previous state-of-the-art model EfficientNet-B0 (Tan & Le (2019)) which trained with AutoAugment (Cubuk et al. (2018)), **1.5%** higher on accuracy with **25M** less FLOPs.

***Resizable-NAS.*** We introduce *ResizableNet for NAS* (Resizable-NAS) that can deploy on many different hardware platforms to fulfill the efficiency constraints with one trained model. Given an efficiency constraint, we can search the specialized scaling configuration of ResizableNet for the target platform. Thus it achieves the run-time model compression objective. Additionally, we observed that there consistently exists smaller sub-network with similar (even higher) accuracy than the baseline network. For example, Resizable-NAS ResNet-50 is **44%** smaller on model size and **0.4%** higher on accuracy compare with the baseline.

***Resizable-Adapt.*** We investigate that if there exists an optimal scaling configuration for every input. We introduce a data-dependent inference network, named as Adaptive ResizableNet (Resizable-Adapt). In our experiment, Resizable-Adapt ResNet-50 improve the performance of Resizable-NAS ResNet-50 up to **0.5%**, depends on the target spatial sub-networks.

## 2 RELATED WORK

**Multi-scale Training.** Our work is most related to multi-scale learning, which is important in image recognition (Li et al. (2019); Wang et al. (2019); Chen et al. (2018)), object detection(Lin et al. (2017); Singh et al. (2018); Singh & Davis (2018)) and instance segmentation (Zhao et al. (2017); Chen et al. (2017b)). There are several approaches to fusing information of images at different visual resolutions. The naive approach is image pyramid (Sermanet et al. (2013); Dalal & Triggs (2005); Felzenszwalb et al. (2009); Cai et al. (2016)), where an input image is passed through a model multiple times at different resolutions. Our work introduce the resizable networks which comprise infinite single scale network, that acted as a implicit multi-scale networks.

**Dynamic Neural Networks.** Our work is related to dynamic neural networks. Dynamic neural networks aim to train a single network to support different architectural configurations, especially adjust networks according to input samples. Slimmable networks (Yu et al. (2018a)) train a single model to support multiple width configurations by switching width at runtime. Universal Slimmable (Yu & Huang (2019)) is proposed to expand the number of width that is supported by training a single network. In our paper, we explore the dynamic network on the spatial level.

**Model Compression.** Compressing Over-parametrized model network with minimal performance drop is appealing on resource-constrained devices. Modern compression methods including channel pruning (Liu et al. (2018); He et al. (2017)), knowledge distillation (Hinton et al. (2015); Furlanello et al. (2018); Romero et al. (2014); Zhang et al. (2018)), hashing (Chen et al. (2015)), network binarization (Courbariaux et al. (2016); Rastegari et al. (2016)). Our work explore network compression at the runtime on spatial-level, which is complementary to existing compression methods.

**Neural Architecture Search.** Recently the design of efficient neural networks has largely shifted from leveraging human knowledge to automatic methods, which is known as neural architecture search (NAS). Designing search space is of the most important in NAS on various vision tasks such as image recognition (Radosavovic et al. (2019); Cai et al. (2018); Howard et al. (2019); Guo et al. (2019); Tan et al. (2019)), instance segmentation (Liu et al. (2019)) and object detection(Ghiasi et al. (2019)). Resizable networks can be consider as a designed search space, where each spatial sub-networks is a optional architecture.

**Data Augmentation.** Data augmentation is effective for improving the accuracy of modern neural network in various computer vision tasks (Szegedy et al. (2015)). Many new augmentation techniques such as Cutout (DeVries & Taylor (2017)) and Mixup (Zhang et al. (2017)) on input image have shown promising results. Recently, a new trend is raised that automatically learn the data transformation (Ho et al. (2019)). Cubuk et al. (2018) leverage RNN controller to automatically search the best data augmentation combination for training in a large augmentation sub-policy space. We explore the data augmentation technique on feature map spatial-level by letting the network learns different feature map scales.

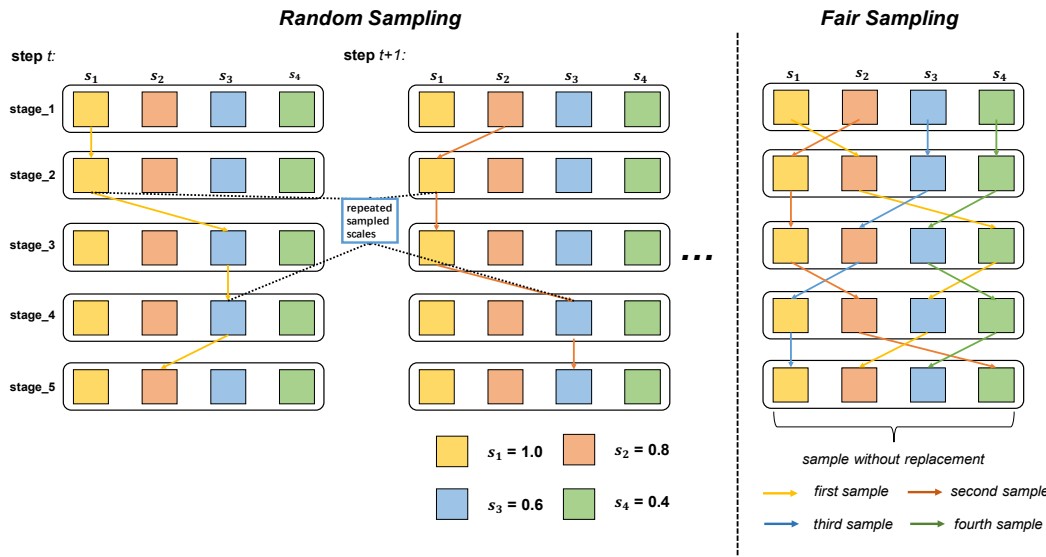

Figure 2: **Fair Sampling**. Left: Random Sampling. In each update step, we sample a sequence of scaling factors at each stage. Right: Fair Sampling. In each update step, we sample a sequence of scaling factors at each stage multiple times. The same scaling factor at the stage does not sample twice in the same update step.

## 3 RESIZABLE NEURAL NETWORKS

Our CNN architecture has $N$ stages. Each stage consists of a sequence of layers that have the same resolution of the feature map. The feature map resolution is gradually reduced by resizing (denoted as $G$) with a *scaling factor* (denoted as $s$), at the beginning of a stage. From the scale perspective, we denote a network by a sequence of resize operation with scaling factors, named as *scaling configuration* (denoted as $S$), denoted formally as

$$S = \{G(s_i)\}_{i=1}^N, i \in N, s_i \in (0, 1],\qquad(1)$$

where $i$ denotes the $i$th stage that performs resize where $i$ denotes the $i$th stage that performs resize operation. In the modern CNN architectures (see Figure 1(a)), $G$ is convolution with stride 2 (He et al. (2016); Huang et al. (2017); Xie et al. (2017)) or max/average pooling (He et al. (2019a); Li et al. (2019)) to reduce the feature map size by half ($s = 0.5$). These CNN architectures learn the input image with a fixed scale. Recent works have indicated the importance of learning the images with various scales. Several works design CNN architecture towards adding parallel branches to deal with different input scales (Li et al. (2019); Wang et al. (2019)). The design of these architectures improve generalization. However, they are inefficient since extra branches require extra parameters and increase FLOPs.

In order to learn image features at different scales without increasing FLOPs, we propose the resizable neural networks. The ResizableNets (see Figure 1c) is a generalization of the CNN architectures in terms of scale. In ResizableNets, $G$ is bilinear interpolation. It allows the feature map to resize to any resolution. As such, modern CNN architecture can be viewed as a special case of ResizableNets. During training, the ResizableNets are trained with different feature map scale dynamically. Moreover, the ResizableNets share the weights of a convolutional layer across different scaling factors. In this way, we allow the network to learn multiple scales of input without extra inference cost.

Training the resizable networks is a non-trivial task due to the flexible resize operation and weight sharing mechanism. We now discuss the inherent problems of ResizableNets and present solutions to resolve these problems.

**Fair Sampling.** The resizable neural network focuses on train one network to support various scaling configurations in inference without retraining. From this perspective, the naive approach is

to randomly sample a scaling factor from a predefined scaling factor list. With a sequence of scaling factors at each stage, we obtain a scaling configuration. We update the weights of ResizableNet use this sampled scaling configuration and repeat this process. However, random sampling brings instability into the ResizableNet training process, because different scaling factor list in a layer are sampled with different number of times by random sampling. As a result, some scaling factors in a convolution have more training opportunities than others, lead to imbalance performance between different scaling configurations. To tackle this problem, we introduce the fair sampling. In each update step, the fair sampling samples a scaling factor from the scaling factor list multiple times. Each time it selects a scaling factor in a layer if and only if this particular scaling factor in this particular layer has not been selected before in the same update step. In this regard, we ensure that all scaling factors in a layer are selected and trained to undergo the same training process. Figure 2 gives an illustration of fair sampling. Moreover, we show in Appendix C that training of ResizableNet with fair sampling is faster than random sampling.

**Scale-Aware Batch Normalization.** Batch normalization (BN) layer is a standard normalization method to accelerate training and improve the generalization of CNN. It is common to use a "*Conv-BN*" couple as a building block. Because the feature map size of convolution is constantly changing during the training of ResizableNets, we say that a BN layer is shared by a convolution layer with multiple sizes. The shared BN layers are not applicable in ResizableNets since different resolutions of the same feature map are aggregated according to Algorithm 1, which results in different mean and variance. It leads to inaccurate batch normalization statistics across different scaling factors in shared layers, based on our experiment in Table 9, it impairs the test accuracy.

To tack this issue, we propose to use an independent BN layer to respond to a scale factor in a layer. In other words, we use a "*Conv-*$\{BN_1, ..., BN_k\}$" as a building block, where $k$ denotes as the number of scaling factors in a layer. As such, each BN layer specifically calculate BN statistics and learns the parameters of corresponding feature map resolution in the stage. The number of extra parameter of scale-aware BN layers is negligible since the number of parameters in BN layers usually take less than one percent of the total model size. During testing, only the BN layer corresponding to the activated scaling factor is used. Thus its behavior is the same as a regular building block.

**Scaling Factor List L.** To help to understand the training process of ResizableNets, we describe scaling factor list L, which we used in Algorithm 1. The scaling factor $s$ for a layer is discrete. It is chosen from a pre-defined scaling factor list $L$. An typical of scaling factor list we used in the experiments is $L = (0.45, 0.65, 0.85, 1.0)$. The number of scaling configurations of ResizableNets is related to the number of stages and the length of $L$ (number of scaling factors). For example, if a network contains four stages, and we use $L = (0.45, 0.65, 0.85, 1.0)$, then there is $4^4 = 256$ scaling configurations. We will discuss the impact of different choice of $s$ and number of $s$ on ResizableNets in Appendix A.

---

**Algorithm 1** Training resizable neural networks RS.

---

**Require:** Initialize resizable network. Define scaling factor list $L$.
**for** $i = 1, ..., n$ **do**
    Get data $x$ and target $y$ of current mini-batch.
    Clear gradients for all parameters, *optimizer.zero_grad()*
    **for** $j = 0, ..., \text{len}(L)$ **do**
        Follows the fair sampling, sample scaling configuration $S_j$,
        Select the scale-aware batch normalization for each stage.
        Execute current scaling configuration. $\hat{y} = S_j(x)$.
        Compute loss for $S_j$, $loss_{s_j}$ = *criterion* $(\hat{y}, y)$
        Compute gradients for $S_j$, $loss_{s_j}$.*backward()*
    **end**
    Update weights, *optimizer.step()*.
**end**

---

# 4 APPLICATIONS

## 4.1 FOR DATA AUGMENTATION

Previous data augmentation techniques transform the input image directly (Zhang et al. (2017); Cubuk et al. (2018); DeVries & Taylor (2017)). In contrast, our method proposes to transform the feature map by scale at stage level, named as Resizable Augmentation (denoted as Resizable-Aug). The network learns various scale combinations of feature map by switching scaling configuration during training, thus potentially improve generalization.

The goal of Resizable-Aug is to increase the accuracy of the target network. The challenge is that a convolutional layer shares the weights across all scaling factors. As a result, training a convolution with a scale could affect the accuracy of the same convolution with other scales. To address this challenge, we decouple the model training stage and the model fine-tuning stage. Precisely, in the model training stage, we follow the training Algorithm 1. As such, the model learns the input with various scales. In the model fine-tuning stage, we specifically fine-tune the target network. In other words, we fix the target scaling configuration and optimize the network with target scaling configuration only. We empirically use two-third of total training epochs at the model training stage and the rest training epochs at the model fine-tuning stage.

## 4.2 FOR NEURAL ARCHITECTURE SEARCH

There is a rising demand for designing specialized neural architecture with a short response time on different hardware (e.g., mobile phones). Recently, neural architecture search methods are proposed to search for the specialized network architecture on a specific hardware. However, searching network architectures for every hardware is computationally expensive, since different hardwares or even the same hardware with different equipment conditions can have different efficiency constraints (e.g., FLOPs and latency).

We present resizable neural networks for neural architecture search (denoted as Resizable-NAS). The ResizableNets can be intuitively viewed as a search space, where scale configurations are candidates in the search space. Every scale configuration of ResizableNets can be instantiated as a sub-network. As such, we can search for the specialized sub-network that fulfill the hardware efficiency constraints. More importantly, with one trained ResizableNets, it can be deployed to diversified hardware simultaneously, thus cut down the computational cost for retraining networks.

To achieve this, we first train a ResizableNet with Algorithm 1. With a trained ResizableNet, we can search for the sub-network for the given efficiency constraints. Specifically, we apply a search method on the ResizableNet to find the specialized sub-networks. For simplicity, we present a naive approach. We enumerate all sub-networks and build a lookup table to include information such as accuracy, FLOPs, and latency. Therefore, given the efficiency constraints, we can query the lookup table to get the specialized sub-network. Additionally, we can increase the accuracy of the specialized sub-network by fine-tuning. Empirically, we find that fine-tune one epoch is sufficient to increase the accuracy considerably, while fine-tune fifteen epochs can obtain the highest accuracy.

## 4.3 FOR DYNAMIC INFERENCE

In our work, we observed that the training accuracy on ImageNet classification for ResizableNets with *smaller scaling configurations* (smaller FLOPs) sometimes perform better than those with *larger scaling configurations*. We conjecture that categories with complex image patterns benefit from predicting with larger scaling configurations, and categories with simpler image patterns benefit from predicting with smaller scaling configurations. (Wang et al. (2019); Li et al. (2019)). Based on this observation, we present dynamic inference for the ResizableNets, named as *Adaptive Resizable Networks* (Resizable-Adapt). Resizable-Adapt aims to find the optimal scaling configuration for each sample based on their inherent characteristics (e.g., object size and image complexity) learned by the network.

For the Resizable-Adapt, we first randomly sample 50k images from the training data. We test all scaling configurations for each sample on an already trained ResizableNet. We take the scale configuration with the highest accuracy for the sample as the label. We then introduce a lightweight

Table 1: This table shows the result of Top-1 accuracy of scaling configurations in resizable networks compare with individually trained counterparts.

| Individual Networks | | | Resizable Networks | | |
|---|---|---|---|---|---|
| Name | FLOPs | Top-1 Acc. | Name | FLOPs | Top-1 Acc.(gain) |
| Individual-MobileNetV1 | 569M | 70.6% | RS-MobileNetV1 | 569M | 72.0%(+1.4%) |
| | 462M | 70.0% | | 462M | 71.3%(+1.3%) |
| | 391M | 69.5% | | 391M | 70.7%(+1.2%) |
| | 291M | 68.1% | | 291M | 69.5%(+1.4%) |
| Individual-ShuffleNetV2 | 299M | 72.6% | RS-ShuffleNetV2 | 299M | 73.3%(+0.7%) |
| | 251M | 71.5% | | 251M | 72.2%(+0.7%) |
| | 199M | 70.3% | | 199M | 70.9%(+0.6%) |
| | 147M | 69.2% | | 147M | 69.8%(+0.6%) |
| Individual-ResNet-50 | 4.1G | 76.4% | RS-ResNet-50 | 4.1G | 77.1%(+0.7%) |
| | 2.8G | 75.4% | | 2.8G | 76.6%(+1.2%) |
| | 2.0G | 74.2% | | 2.0G | 75.2%(+1.0%) |
| | 1.1G | 71.6% | | 1.1G | 72.3%(+0.7%) |

scaling configuration classifier that is attached at the last stage of the original resizable network. It is composed of three convolutional layers along with a pooling layer and followed by the softmax. It learns the image specific scaling configuration. To be specific, this classifier takes an image as input and predicts an optimal scaling configuration for this input. Then we switch the ResizableNet to this scaling configuration and predict its labels using the input.

Our extensive experiments verify our conjecture that the network can indeed achieve better performance if the sample is predicted by a suitable scaling configuration. Additionally, the previous resizable network process all images to one scaling configuration, as the network spends equal energy (e.g., FLOPs) at every sample. The adaptive resizable network can spend equal or less amount of computational cost during testing since some samples are assigned to smaller-scaling configurations. Thus, Resizable-Adapt is more efficient in inference.

## 5 EXPERIMENTS

In this section, we validate the ResizableNets on ImageNet classification from several perspectives. Implementation details can be found in the Appendix A.1. Section 5.1 give experiments for resizable networks compared with individual networks. Section 5.2, Section 5.3, and Section 5.4 gives the result of Resizable-NAS, Resizable-Aug and Resizable-Adapt network respectively.

**Notation.** For consistency, we use *RS-* to indicate ResizableNets from Section 3, *RS-NAS* indicates Resizable-NAS from Section 4.2, *RS-Aug* to represent resizable augmentation from Section 4.1. *RS-Adapt* to denote adaptive resizable networks from Section 4.3.

### 5.1 COMPARISON BETWEEN RESIZABLE NETWORKS WITH STAND-ALONE NETWORK.

We show in Table 1 the top-1 classification accuracy for both individually trained networks and resizable networks give the same *scaling configurations*. Due to the space limit, we only take several scaling configurations for comparison. We present top-1 test accuracy and FLOPs, which is the primary concerns in inference time, of all networks for reference. Compared to independently trained networks, the spatial sub-networks in resizable networks achieved significantly better top-1 accuracy, across various network architectures and computational complexity. This result confirm our hypothesis that the implicit multi-scale training improve the model capacity.

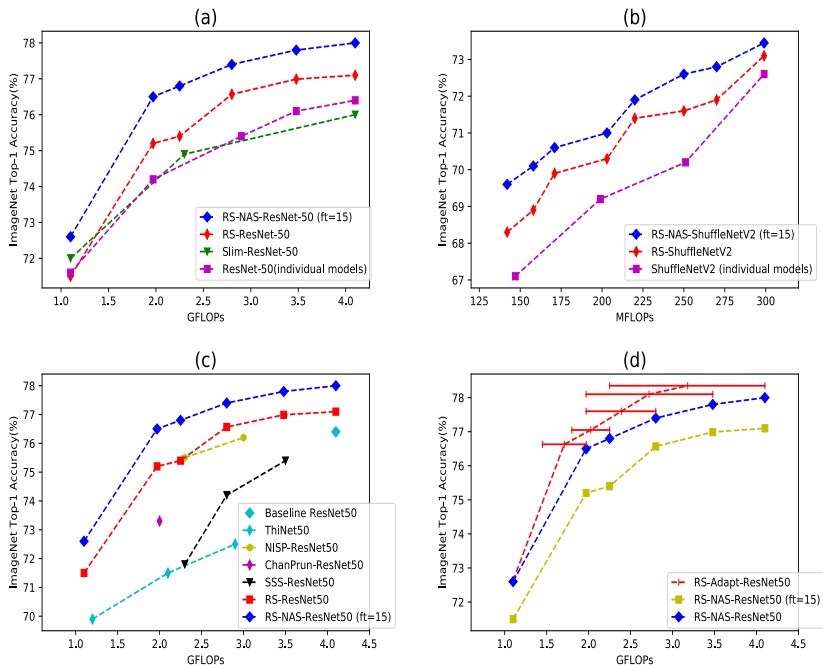

Figure 3: (a)-(b): Resizable-NAS ResNet-50 and ShuffleNetV2 compare with the individually trained networks and naive resizable approach. (c): Resizable-NAS ResNet-50 compare with state-of-the-art model compression methods. (d) Resizable-Adapt ResNet-50 compare with the Resizable-NAS ResNet-50. The horizontal line mean the FLOPs range of sub-network used in resizable-adapt network for data-dependent inference. Models with larger target network achieve better accuracy.

Table 2: This table summarizes the validation Top-1 accuracy on ImageNet dataset for ResNet-50 architecture. We compare Resizable-Aug with multiple state-of-the-art augmentation methods and regularization methods. Bold indicates the best results. [*] denotes results reported in the original papers.

| Model | Top-1 Acc.(gain) | Top-5 Acc.(gain) |
|---|---|---|
| Baseline ResNet-50 (He et al. (2016)) | 76.4% | 93.1% |
| ResNet-50 + Cutout (DeVries & Taylor (2017)) | 76.5% | 93.2% |
| ResNet-50 + MixUp[*] (Zhang et al. (2017)) | 76.7% | 93.4% |
| ResNet-50 + AutoAugment[*] (Cubuk et al. (2018)) | 77.6% | 93.8% |
| **ResNet-50 + RS-Aug** | **78.8%**(+2.4%) | **94.4%**(+1.3%) |

## 5.2 Resizable-NAS.

**Comparison with Naive Resizable Networks.** The Figure 3(a) and 3(b) demonstrate the results of ResizableNet and Resizable-NAS network with ResNet-50 and ShuffleNetV2. The only difference between these two networks is that the Resizable-NAS network take 15 epochs to fine-tune on target scaling configuration. The Resizable-NAS shows the notable performance improvement over individual network and ResizableNet with the same scaling configuration.

**Comparison with State-of-the-art Model Compression Methods.** We compare with the following state-of-the-art model compression methods: NISP (Yu et al. (2018b)), ThiNet (Luo et al. (2017)), ChannelPruning (He et al. (2017)), Sparse Structure Selection ResNet50 (Huang & Wang (2018)) and SlimmableNets (Yu et al. (2018a)), AMC (He et al., 2018) and Netadapt (Yang et al., 2018). We

Table 3: Comparison with state-of-the-art architectures on ImageNet (200M to 400M FLOPs). [*] denotes reported results employ AutoAugment.

| Model | FLOPs (M) | Params (M) | Top-1 Acc. | Top-5 Acc. |
|---|---|---|---|---|
| ShuffleNetV2 1.5x (Ma et al., 2018) | 300 | - | 72.6% | - |
| MobileNetV2 1.0x (Sandler et al., 2018) | 300 | 3.4 | 72.0% | 91.0% |
| MobileNetV3 Large (Howard et al., 2019) | 291 | 5.4 | 75.2% | 92.2% |
| MnasNet-A2 (Tan et al. (2019)) | 340 | 4.8 | 75.6% | 92.7% |
| EfficientNet-B0[*] (Tan & Le (2019)) | 390 | 5.3 | 76.3% | 93.2% |
| Efficient ResizableNet w/o Aug | 365 | 6.7 | 77.4% | 93.6% |
| Efficient ResizableNet | 365 | 6.7 | 77.8% | 93.6% |

present the ResNet-50 based results at Figure 3(c), and the detailed results of Resizable-NAS MobileNetV1 as well as Resizable-NAS ResNet-50 can be found in Table B. Resizable-NAS ResNet-50 demonstrate strong performance, especially when the compression ratio is large than 50%. It outperforms listed state-of-the-arts compression methods by a large margin over diverse computational complexity constraints. We emphasize that our Resizable-NAS network only trained once, and fine-tune 15 epochs on target sub-networks, where the other compression methods find the optimal structure and retrain accordingly. Comparing to the other methods, our proposed model is more efficient while performing better. Notice even the version without the fine-tuning process can achieve similar (even better) performance than other model compression methods.

## 5.3 RESIZABLE AUGMENTATION

**On ResNet-50.** We compares the Resizable-Aug ResNet-50 with various state-of-the-art data augmentation methods (i.e., Cutout (DeVries & Taylor (2017)), Mixup (Zhang et al. (2017)) and AutoAugment (Cubuk et al. (2018))). The results are summarized in Table 2. It shows that Resizable-Aug improve the ResNet-50 baseline by 2.4%, outperform AutoAugment by 1.2% using the same the baseline. It is worthy to note that our Resizable-Aug takes the same number of epochs as the baseline to train the network. In our experiment, we use scaling factor list of $(0.45, 0.65, 0.85, 1.0)$.

**On Efficient Models.** We present the resizable augmentation results based on network that are search by AutoML. The details of the training strategy and network architecture are presented in the Appendix B.3. Table 3 shows the results. By adopting the resizable augmentation strategy, we improve the already high accuracy network from 77.4% to 77.8%, achieve a new state-of-the-art result. The new models is 1.5% better than EfficientNet-B0 (Tan & Le (2019)), which is trained AutoAugment (Cubuk et al., 2018) with 25M less FLOPs. In our experiment, we use scaling factor list of $(0.45, 0.65, 0.85, 1.0)$.

## 5.4 ADAPTIVE RESIZABLE NEURAL NETWORK

Figure 3(d) shows the results of Resizable-Adapt ResNet-50. The dynamic inference is performed on multiple computation complexity threshold. During inference, we ensure that only sub-networks that are smaller than the target network is used for inference. The performance gain is range from 0% to 0.5%. It can be observed that the adaptive resizable ResNet-50 consistently obtain better performance than the baseline. This is a surprising result since most dynamic network struggle at the trade-off between efficiency and accuracy. Less computational complexity in dynamic network always lowers the accuracy compared with the baseline. Moreover, when the target network is large, sometimes the samples are still better predicted by the small sub-network. This gives the evidence that the explicit way to deal with the input scale is indeed beneficial.

## 6 CONCLUSION

We introduced a type of convolutional neural network, named as Resizable Neural Networks. The resizable networks can do arbitrary resize operation on feature map at stage level. We further provide essential tools to build and train our proposed network. Moreover, we give three applications

of ResizableNet. Specifically, we present Resizable-NAS, Resizable-Aug, and Resizable-Adapt. Resizable-Aug is a data augmentation techniques which do feature map transform by scaling and achieve state-of-the-art results on ImageNet classification. Resizable-NAS can deploy on many hardwares with various efficiency constraints by on trained network. Resizable-Adapt performs data-dependent dynamic inference. Overall, we believe that the resizable network's approach provides an interesting and practical new perspective on designing a convolution neural network on the spatial dimension.

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

## A CHOOSING THE PREDEFINED SCALING FACTOR LIST.

We will discuss the choice of predefined scaling factor list to ResizableNet and ResizableAug separately.

### A.1 RESIZABLENET

We first report the scaling configuration list we used in the previous experiments. For all Resizable ResNet-50 and Resizable ShuffleNetV2, we use scaling configuration $S = (0.45, 0.65, 0.85, 1.0)$. For all Resizable MobileNetV1 we use $S = (0.5, 0.7, 1.0)$. We use only three scaling factors because the number of stage of MobileNetV1 is one larger than ResNet and ShuffleNetV2.

**Choice of Scaling Configuration *S*.** The choice of scaling configuration depends on the range of efficiency constraints that need to be covered. There is no specific guideline for choosing scaling configuration for ResizableNet.

**Number of scaling factors *n*.** With more scaling factors, the number of scaling configuration are increased exponentially, thus more flexible during testing. We investigate how the number of scaling factors impact accuracy on Resizable-NAS. In Table 4, we train ResNet-50 with two different scaling factor list. We compare a four scaling factors list with a six scaling factors list. We display some results. The results of two different number of scaling factors are similar, showing the scalability of our proposed method.

Table 4: This table shows the result of Top-1 accuracy of scaling configurations in resizable networks compare with individually trained counterparts.

| Scaling Factors | FLOPs | Top-1 Acc. | Scaling Factors | FLOPs | Top-1 Acc. |
|---|---|---|---|---|---|
| (0.45, 0.65, 0.85, 1.0) | 3.5G | 77.8% | (0.45, 0.55, 0.65, 0.75, 0.85, 1.0) | 3.5G | 77.5% |
| | 2.8G | 77.4% | | 2.8G | 77.1% |
| | 2.3G | 76.8% | | 2.3G | 77.0% |
| | 2.0G | 76.5% | | 2.0G | 75.8% |
| | 1.0G | 72.8% | | 1.0G | 72.9% |

### A.2 RESIZABLEAUG

For all ResizableAug experiment, we use scaling configurations of $S = \{0.45, 0.65, 0.85, 1.0\}$.

**Choices of scaling configuration *S*.** We study how to choose the scaling factors set $S$ such that the best performance via ResizableAug can be achieved. The performance gain in ResizableAug is due to the implicit multi-scale training. Thus the predefined scaling factor list can be viewed as the choice for multi-scale. We empirically find that, as long as we use the diversified scaling factor, the performance of ResizableAug is robust to the specific choice of scaling factors. The left part of Table 6 shows the results using different combinations of scaling factors. The first three experiments, $S = (0.45, 0.65, 0.85, 1.0)$, $S = (0.4, 0.6, 0.8, 1.0)$ or $S = (0.25, 0.5, 0.75, 1.0)$ achieve the same or similar performance, which indicate the specific choice of scaling factor is a critical to increase the accuracy of the model. In the fourth set, we use $S = (0.7, 0.8, 0.9, 1.0)$, where every scaling factor is close to each other, and the accuracy falls as we expected. By default, in all our experiment we use $S = (0.45, 0.65, 0.85, 1.0)$.

**Number of scaling factors *n*.** We study how the number of scaling factor candidates affect the performance of ResizableAug. We train three ResizableAug ResNet-50 with number of scaling factors equal to 3, 4 or 5, following the scaling factor diversity rule. Right part of Table 6 shows that the model trained with $n = 4$ or $n = 5$ achieve the same performance, while $n = 3$ perform worse than $n = 4$ or $n = 5$. By default, in all our experiment we use $n = 4$

Table 5: This table shows the result of Top-1 accuracy of ResizableAug ResNet-50 by various scaling configuration.

| Length of Scaling Config. | | Choice of Scaling Config. | |
|---|---|---|---|
| Scaling Config. | Acc. | Scaling Config. | Acc. |
| (0.45,0.65,0.85,1.0) | 78.8% | (0.45,0.65,0.85,1.0) | 78.8% |
| (0.4,0.6,0.8,1.0) | 78.8% | (0.25,0.45,0.65,0.85,1.0) | 78.8% |
| (0.25,0.5,0.75,1.0) | 78.7% | (0.5,0.75,1.0) | 77.6% |
| (0.7,0.8,0.9,1.0) | 77.9% | | |

Table 6: This table shows the result of Top-1 accuracy of ResizableAug ResNet-50 by various scaling configuration.

## B    IMPLEMENTATION DETAILS

### B.1    NETWORK IMPLEMENTATION DETAILS.

Resize operation is place at the first non-stride block at each stage in network. We apply bilinear interpolation to resize the feature map to target size. Resize factor are uniformly taken from a predefined list of possible resize choice, except when it contradicted to fair sampling rule. Our implementation is based on the Pytorch (Paszke et al. (2017)). The input resolution for all train and test images for ImageNet classification are 224 x 224.

Other than ResNet-50, which we report baseline higher than the original paper, we have trained Mobilenet v1, Mobilenet v2 and Shufflenet v2 baseline to match the performance as reported in original paper. And based on the re-implemented settings, we perform both baseline and resizable networks experiments.

For ResNet-50, we train for 100 epochs using weight decay of 1e-4. We use batch size of 512 and do learning rate decay by 10 at epochs 30, 60, 80, 90. The other resizable network train for 200 epochs and double the epochs of learning rate decay.

For Mobilenet v1, Mobilenet v2 and Shufflenet v2, we use batch size of 1024 and weight decay of 4e-5. We train both models for 240 epochs using linear learning rate decay where base learning rate is set to 0.5. All models are train across synchronous SGD across 8 GPUs. The resizable networks train for 480 epochs

The resizable augmentation network always use the same training setting as the baseline method. Extending the training epochs is unnecessary.

### B.2    DETAIL RESULTS OF RESIZABLE-NAS NETWORKS

The detail results of Resizable-NAS ResNet-50 and Resizable-NAS MobileNetV1 can be found at Table B.

### B.3    DETAILS OF EFFICIENT RESIZABLENET

**Training Details.** Efficient ResizableNet is trained using resizable-aug technique as we discuss in the Section 4.1. It is trained for 480 epochs with batch size 1024 on 8 GPUs. The network parameters are optimized using an SGD optimizer with an initial learning rate of 0.5 (decayed linearly after each iteration), a momentum of 0.9 and a weight decay of $3 \times 10^{-5}$. Additional enhancements including label smoothing (Szegedy et al. (2016)) and Dropout with 0.5 on last FC layer. We follow the common practice to use inception augmentation. This architecture is searched with Anonymous (2020) using search space proposed by Guo et al. (2019). The Xception is proposed by Chollet (2017).

**Detailed Network Architecture.** The network architecture along with feature map resolution and channels number are shown in Figure B.3.

| Model | Top-1 Acc. | Gain | FLOP | Compres. Ratio |
|---|---|---|---|---|
| ResNet50 (He et al. (2016)) | 76.4% | +0% | 4.1G | 0% |
| **RS-NAS-ResNet50 (ft=15)** | **77.8%** | **+1.4%** | **3.5G** | **15%** |
| RS-NAS-ResNet50 (ft=1) | 77.6% | +1.2% | 3.5G | 15% |
| RS-ResNet50 | 77.1% | +0.7% | 3.5G | 15% |
| **RS-NAS-ResNet50 (ft=15)** | **77.4%** | **+1.0%** | **2.8G** | **32%** |
| RS-NAS-ResNet50 (ft=1) | 77.1% | +0.7% | 2.8G | 32% |
| RS-ResNet50 | 76.6% | +0.2% | 2.8G | 32% |
| SSS-ResNet50 (Huang & Wang (2018)) | 74.2% | -2.2% | 2.8G | 32% |
| ThiNet-70 (Luo et al. (2017)) | 72.5% | -3.9% | 2.9G | 29% |
| NISP-ResNet50-A (Yu et al. (2018b)) | 76.2% | -0.2% | 3.0G | 27% |
| **RS-NAS-ResNet50 (ft=15)** | **76.8%** | **+0.4%** | **2.3G** | **44%** |
| RS-NAS-ResNet50 (ft=1) | 76.2% | -0.2% | 2.3G | 44% |
| RS-ResNet50 | 75.4% | -1.0% | 2.3G | 44% |
| Slim ResNet50 (Yu et al. (2018a)) | 75.0% | -1.4% | 2.3G | 44% |
| SSS-ResNet50 (Huang & Wang (2018)) | 75.6% | -0.8% | 2.3G | 44% |
| NISP-ResNet50-B (Yu et al. (2018b)) | 75.5% | -0.9% | 2.3G | 44% |
| **RS-NAS-ResNet50 (ft=15)** | **76.5%** | **+0.1%** | **2.0G** | **51%** |
| RS-NAS-ResNet50 (ft=1) | 75.9% | -0.5% | 2.0G | 51% |
| RS-ResNet50 | 75.2% | -1.2% | 2.0G | 51% |
| ChanPrun-ResNet50 (He et al. (2017)) | 73.3% | -3.1% | 2.0G | 51% |
| ThiNet-50 (Luo et al. (2017)) | 72.5% | -3.9% | 2.1G | 49% |
| **RS-NAS-ResNet50 (ft=15)** | **72.8%** | **-3.6%** | **1.1G** | **73%** |
| RS-NAS-ResNet50 (ft=1) | 72.4% | -4.0% | 1.1G | 73% |
| RS-ResNet50 | 72.3% | -4.1% | 1.1G | 73% |
| Slim-ResNet50 (Yu et al. (2018a)) | 72.0% | -4.4% | 1.1G | 73% |
| ThiNet-30 (Luo et al. (2017)) | 69.9% | -6.5% | 1.2G | 71% |
| MobileNetV1 (Howard et al. (2017)) | 70.9% | +0% | 569M | 0% |
| **RS-NAS-MobileNetV1 (ft=15)** | **71.9%** | **+1.0%** | **391M** | **31%** |
| **RS-NAS-MobileNetV1 (ft=15)** | **70.9%** | **+0%** | **291M** | **49%** |
| RS-MobileNetV1 | 69.5% | -1.4% | 291M | 49% |
| Slim-MobileNetV1 (Yu et al. (2018a)) | 69.5% | -1.4% | 317M | 44% |
| NetAdapt-MobileNetV1 (Yang et al. (2018)) | 70.1% | -0.9% | 285M | 50% |
| AMC-MobileNetV1 (He et al. (2018)) | 70.5% | -0.4% | 285M | 50% |

Table 7: This table compares the Top-1 accuracy of Resizable Network with several state-of-the-art model compression methods given FLOPS. Bold indicates highest top-1 accuracy under certain FLOPS constraint. Blue indicates dynamic networks.

## C   TRAINING EFFICIENCY ANALYSIS.

We analysis the training time of ResizableNets. We evaluate the training times of ResizableNets with various number of scaling factors (1, 2, 4, and 6) are evaluated, as well as the total training time if the same amount of sub-networks in the resizable net are trained individually. The results are shown in Table 8. On the one hand, the total training time of resizable nets is remarkably less than the train the same amount of individual networks separately due to the weight sharing mechanism. On the other hand, because we sample multiple scaling configurations in one step, we eliminate the need to provide mini-batch data and update gradients repeatedly. As a result, the training time by fair sampling is less than simply extending the total training epochs by the number of predefined scaling factors. Particularly, as the number of scaling factors increases, the benefit of fair sampling in terms of training time is more significant. For example, compare to random sampling, when $len(S) = 4$, fair sampling is 10% faster, while $len(S) = 8$, fair sampling is 52% faster.

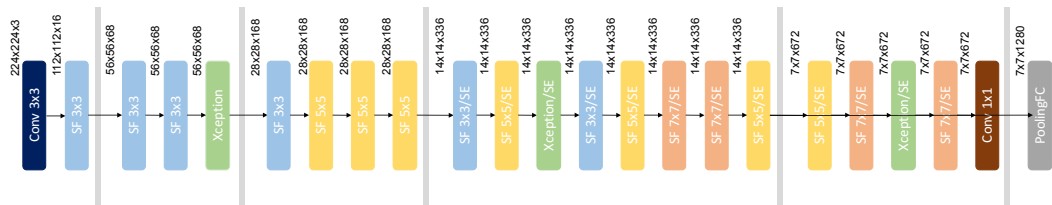

Figure 4: Architecture of ResizableNet. We highlight the input and output tensor shape. Conv denotes convolution layer. SF denotes ShuffleNetV2 module. SE denotes Squeeze-and-Excitation module.

| Length | # of Networks | Training time (gain.) | | |
| --- | --- | --- | --- | --- |
| | | Individual | Random | Fair |
| 1 | 1 | 8 Gds | 1 | 8 Gds (+0%) |
| 4 | 625 | 4,896 Gds* | 32 Gds* | 29 Gds (+10%) |
| 6 | 15625 | 122,395 Gds* | 48 Gds* | 35 Gds (+26%) |
| 8 | 390,625 | 3,125,000 Gds* | 64 Gds* | 42 Gds (+52%) |

Table 8: Training time comparison among training individual networks, ResizableNet with random sampling and fair sampling, with different number of scaling factors in the predefined scale list. It runs by algorithm 1 use ResNet-50 on ImageNet with Pytorch on 8 Nvidia 2080 Ti GPUs with 100 epochs. Gds: GPUs per day. * denotes the approximate training time. Number in the bracket represents the percentage of training time gain of fair sampling compare to random sampling.

# D    ABLATION STUDY

## D.1    IS NEURAL NETWORK NATURALLY RESIZABLE?

It is intuitive to ask the question: is neural network naturally resizable? The answer is no. We present the result of directly inference on a neural network with different scaling configurations and compare the result with resizable neural network. The naive network is trained with scale-aware batch normalization. Figure D.1 shows that naively trained neural network can not adjust feature map size, which result in a extremely low test accuracy.

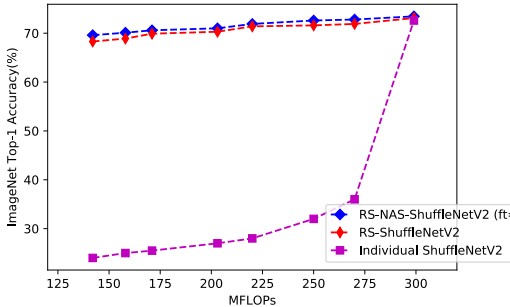

Figure 5: **Is network naturally resizable?** We compare the result of sub-networks in naive ShuffleNetV2, RS-ShuffleNetV2 and RS-NAS-ShuffleNetV2.

## D.2    SCALE-AWARE BN ANALYSIS.

Scale aware batch normalization is especially important for resizable augmentation. The following table compares two networks to adopt resizable augmentation with and without BN calibration. We

can observe that without BN calibration, the resizable augmentation is only slightly better than the baseline model. The improvement is negligible. On the other hand, when BN calibrated for scaling configurations, is boost the performance compare with the baseline. The accuracy is increased by 1.1% on ShuffleNetV2 and 2.4% on ResNet-50.

Table 9: This table shows the result of Top-1 accuracy of ResNet-50 and ShuffleNetV2 use resizable augmentation technique with (w/) and without (w/o spatial BN calibration.)

| Model | Baseline | w/o BN Calibration | w/ BN Calibration |
|---|---|---|---|
| Resizable-Aug ShufflenetV2 | 72.6% | 72.8% | 73.7% |
| Resizable-Aug ResNet-50 | 76.4% | 76.9% | 78.8% |

### D.3   FAIR SAMPLING ANALYSIS.

We compare the results of random sampling and fair sampling in this section. We use the scaling factor list $L = (0.5, 0.75, 1.0)$ on MobileNetV1. The random sampling is trained with three times more epochs than fair sampling. We showed some results in Table 11. The results show that the performance of different scaling configuration is not consistent. By consistency, we mean that some scaling configuration achieves better accuracy, while some scaling configuration achieves lower accuracy, using fair sampling we can get more stable results.

Table 10: This table shows the result of Top-1 accuracy of ResizableAug ResNet-50 by various scaling configuration.

| Random Sampling | | Fair Sampling. | |
|---|---|---|---|
| FLOPs | Acc. | FLOPs | Acc. |
| 569M | 72.0% | 569M | 71.9% (-0.1%) |
| 462M | 71.3% | 462M | 69.3% (-2.0%) |
| 391M | 70.7% | 391M | 71.0% (+0.3%) |
| 343M | 70.2% | 343M | 70.7% (+0.5%) |
| 291M | 69.5% | 291M | 65.7% (-3.7%) |
| 242M | 68.9% | 242M | 68.1% (-0.8%) |

Table 11: This table shows the result of Top-1 accuracy of ResizableAug ResNet-50 by various scaling configuration.

