# OpenReview forum: "Resizable Neural Networks"
_ICLR.cc/2020/Conference — Reject_

### Official Review · AnonReviewer2 · 2019-10-07
**Official Blind Review #2**

**Rating:** 6

**Review:**

In a standard multi-stage neural network such as a ResNet, the resize operation between stages typically reduces the spatial resolution by 0.5.  (input ---> stage 1---> 0.5 reduction ---> stage 2 --> 0.5 reduction ---> ....). In this paper the authors apply a variety of reduction factors in between these stages for different training examples (input ---> stage 1---> variable reduction ---> stage 2 --> variable reduction ---> ....). They demonstrate that simply training in this way is a powerful form of augmentation. It also produces a network which may be scaled depending on a FLOP budget.

I think this is a really neat idea, and as far as I'm aware it is novel.  It is similar in spirit to EfficientNet, although more flexible. The experimental results are good. However, the paper is let down by poor writing and a lack of detail.

The paper needs a rewrite as there are many grammatical errors, which cause a bad impression:

- "performs arbitrary resize operation" ->  "performs arbitrary resize operations"
- "Scale is the fundamental component of the physical world" ---> *a* fundamental component!
- "i.e the acuracy of NISP" --> e.g.
- "compare with baseline" -> "compared to the baseline"
- There are several instances of a space missed out between a letter and an open bracket
- "The Figure 2(a)" --> "Figure 2(a)"

The comparison to weight-sharing NAS methods is unnecessary. Those entail searching for architectures and sharing weights through the search process, whereas in this work it is having a network that can take different sized inputs at different stages. On that note, it doesn't feel right to me to refer to there being many "subnetworks". What there really is, is just a single network that is robust to multi-scaled inputs (which is a good thing!)

Figure 1 is nice!

EfficientNet-B0 is a base-network that can be scaled up with a compound scaling approach found through grid search. What happens if you scale up your Resizeable net to the same FLOPs as e.g. EfficientNet-B7?

I like the old-school vision citations, although referring to object detection makes me wonder why there are no experiments on it. For distillation, I recommend you cite https://arxiv.org/abs/1312.6184, as the Hinton paper is really just an extension of this.

The fair sampling seems important to the method. Could a detailed explanation be included in the appendix? Do you given the performance as a result of naive sampling? What do you mean by "Some convolutional layers can go through more training issues than others"?

A big problem in this paper is that (as far as I can tell) the scaling factors considered aren't given (but we are told that they lie between 0 and 1). It isn't possible to sample these arbitrarily as you indicate that a batch-norm layer is needed for each one, so these must be discrete (because of this, it isn't correct to say that you have infinite networks contained within).  It therefore isn't clear e.g. in Table 1 which permutation of scaling factors were used in upsampling the networks. Are the results given representative of the best-case selection of these scaling? I hope this isn't the case. I am assuming it is uniform scaling in the case of the "individually trained counterparts". It would be interesting to know more generally which combinations worked well.

In Section 3.5 I'm not sure what is meant by "It is inefficient to process all images to one-subnetwork, as the algorithm spends equal energy at each sample".  I assume energy use is proportional to the number of FLOPS, which in turn depends on spatial resolution.

The legend in Figure 2 is close-to unreadable and needs changing.

The results are impressive, but error bars would be appreciated if possible. As ImageNet is the only dataset considered, this would give some needed clout.

Pros
-------

- Nice, novel method
- Good experimental results

Cons
--------

- Paper is poorly written
- Very few details of the scaling factor variations
- Only one dataset considered

Although the paper is written badly and the narrative is muddled, the underlying idea is a nice one, which is executed well experimentally. Because of this I would like to recommend a Weak Accept, subject to the authors (i) doing a rewrite and (ii) including more information regarding the scaling permutations.

**Experience Assessment:**

I have published one or two papers in this area.

**Review Assessment: Checking Correctness Of Derivations And Theory:**

N/A

**Review Assessment: Checking Correctness Of Experiments:**

I assessed the sensibility of the experiments.

**Review Assessment: Thoroughness In Paper Reading:**

I read the paper thoroughly.

---

> ### Author Response · Authors · 2019-11-15
> **Response to Reviewer #2**
>
> Reviewer 3:
> Thanks for your detailed and thoughtful reviews as well as your helpful suggestions.
>
> Q1: Regarding writing, grammatical error, typos.
> A1: We improved the writing and fixed the typos. We enlarged Figure 2 for readability. As you suggested, we weaken the idea of sub-networks. The link of distillation paper you mentioned is out-dated, we will include this in our reference if you can kindly provide the name of the paper.
>
> Q2: Scale up resizable network?
> A2: The resizable network can indeed scale up. We did preliminary experiments on scale up ResNet-50 by 4x times larger FLOPs and achieve better performance we reported in EfficientNet paper (Table 3). We will consider your advice and do more experiments to see the possibility of scaling up the resizable network.
>
> Q3: More details and ablation study on fair sampling.
> A3: We improved writing and included more descriptions of fair sampling. We added an ablation study on fair sampling.
>
> Q4: Regarding the scaling factor combination.
> A4: We included the choice of scaling factor combination we used in the experiments section. We added a new section in Appendix A to discuss the impact of the choice of scaling factor on our method.
>
> Q5: In Section 3.5 I'm not sure what is meant by "It is inefficient to process all images to one-subnetwork, as the algorithm spends equal energy at each sample".  I assume energy use is proportional to the number of FLOPS, which in turn depends on spatial resolution.
> A5: Yes. We understand the description of energy can be ambiguous; thus, we make a revision on Section 3.5 (Section 4.3 in new revision).
>
> Q6: As ImageNet is the only dataset considered in the paper, could you provide error bars?
> A6: At this time, we can not provide error bars due to limited time constraints. We will consider your advice to provide error bars and do more experiments on other datasets (other tasks) in the future revision.

---

### Official Review · AnonReviewer3 · 2019-10-23
**Official Blind Review #3**

**Rating:** 3

**Review:**

This paper proposes Resizable Neural Networks, which trains networks with different resolution scalings at the same time with shared weights. It serves as data-augmentation and improves accuracy over base networks. Additionally, the same technique can perform an architecture search. Experimental results show significant accuracy gains.

The reported accuracy gains are substantial. The proposed method is potentially useful in many applications. However, several details are missing or hard to understand. Without additional descriptions, it is not straightforward to implement the method. Thus, I suggest for rejection. The score might be raised depending on updates and code release.

Major comments:
1) Algorithm 1 has "predefined spatial list L." How to choose it in practice?
2) Algorithm 1 indicates that the training time is len(L) times longer. Additionally, according to the implementation details, Resizable Networks are trained two times many epochs. It seems hard to justify such a longer training time.
3) This is similar to (2), but why Resizable-NAS is better than the all len(L) models separately to find better architectures?
4) In Sec. 3.4, how the "target model" is selected after the training of Resizable Networks?
5) Why is Resizable-Adapt omitted from Table 2?
6) References are out-dated. For example, there are no "Figure 5."

Minor comments:
1) Are any ablation studies available for Fair Sampling?

===== Update

Thank you for the response and update. The revision made the paper easier to understand. However, I still have concerns about the presentation of the paper, and I keep my score. I think the novelty and experimental results are significant and match the bar of ICLR. If the other two reviewers think that the revised paper is fairly well-written and recommend acceptance, I will not challenge the decision.

Major comments:
1) I do not understand how the training times of random-sampling in Appendix C were estimated.
2) Concerning Table 11, how was the result when we randomly select each scale and train network with the same number of epochs with fair-sampling? If there were no much difference with the result in Table 11, it seems fair sampling does not have clear advantages over random sampling. If so, I suggest to alter fiar-sampling by random-sampling and reduce the complexity of the proposed method.

Minor comments:
1) Table 10 has two captions.
2) I did not understand that L is a list of possible scaling factors (scalar) in the initial review. I guess it is partially because I did not understand why the number of sampling should be len(L) for each mini-batch (it is actually due to fair-sampling). I think adding some remarks on it will help to make the pseudo-code easier to understand. (nit: for j= 0, ..., len(L) should be for j=0, ..., len(L) -1 or for j=1,  ..., len(L))

**Experience Assessment:**

I have read many papers in this area.

**Review Assessment: Checking Correctness Of Derivations And Theory:**

N/A

**Review Assessment: Checking Correctness Of Experiments:**

I assessed the sensibility of the experiments.

**Review Assessment: Thoroughness In Paper Reading:**

I read the paper at least twice and used my best judgement in assessing the paper.

---

> ### Author Response · Authors · 2019-11-15
> **Response to Reviewer #3**
>
> Thanks for your detailed and thoughtful reviews. We have addressed all the questions below:
>
> Q1: Implementations details and code release.
> A1: We understand that in our previous version, several details are missing, and some concepts are hard to understand. Besides improving writing and fixing the typos, we included more details (both method descriptions and experiment settings) in the new revision. We are working on cleaning the code, and we will soon release the code.
>
> Q2: Algorithm 1 has "predefined spatial list L." How to choose it in practice?
> A2: We added a subsection in method (Section 3) to define L. We included the choice of L we used in the experiments section. We added a new section in Appendix A to discuss the choice of pre-defined spatial list L.
>
> Q3: How to justify a longer training time for resizable network.
> A3: Regarding your question on training time, we claim that the resizable network is a single trained network that can scale depending on a FLOPs budget. The ResizableNets indeed need more training time, because it is trained with multiple scale configurations in one network.  On the one hand, it is more efficient than training hundreds of individual networks with different feature map scale configuration. On the other hand, if we have a target network and use resizable augmentation to improve generalization in Section 3.4 (Section 4.1 in the new revision), the training time is less or equal to other data augmentation techniques. Moreover, the performance gain of ResizableNet-NAS is not coming from longer training time. We did experiments to increase the training time of ShuffletNetV2, and the result is below:
> ShuffletNetV2 1.5x:
> Baseline (250 epochs): 72.6%
> 2x longer training time (500 epochs): 72.6%
> 8x longer training time (2000 epochs: 72.6%
> Our preliminary experiment indicates that simply extend the training time can not improve the performance of the network.
>
>
> Q4: Why Resizable-NAS is better than the all len(L) models separately to find better architectures?
> A4: We do not claim that it is better to use Resizable-NAS for finding better architectures compared to training all len(L) models separately. Admittedly, training all models separately and then find the network in these already trained models would be the best option. However, the training cost is overwhelming. Resizable-NAS is more efficient because we train different scale configuration in a single network with shared weights, then search for the target sub-network that satisfy the efficiency constraints.
> More importantly, our contribution of Resizable-NAS is not the searching method since we propose to use a lookup table, which is not the most efficient way for searching (other searching methods such as evolution algorithm can be more efficient). Instead, our contribution is that we propose an application scenario for ResizableNet. As far as we are aware, our paper is the first do neural architecture search on **resolution** and showed surprisingly good results.
>
> Q5: How the "target model" is selected after the training of Resizable Networks?
> A5: The target model is predefined. It depends on which scale configuration you wish to apply the resizable augmentation technique. The experiments on resizable augmentation in our paper use the same setting as the baseline methods (all scale factors in the scale configuration is 0.5; s = 0.5).
>
> Q6: Why is Resizable-Adapt omitted from Table 2?
> A6: The Resizable-Adapt is a dynamic inference method. Since Table 2 compared the performance of different data augmentation techniques, we don't think it is fair to include Resizable-Aug in Table 2.
>
> Q7:References are out-dated. For example, there are no "Figure 5."
> A7: We fixed it in the new revision.
>
> Q8: Ablation studies for Fair Sampling?
> A8: We included ablation studies on Fair Sampling in the Appendix.

---

### Official Review · AnonReviewer1 · 2019-10-25
**Official Blind Review #1**

**Rating:** 3

**Review:**

This paper proposes a new method that involves multi-scale inputs for each layer that could be used as network architecture search or data agumentation or

Pros)
(+) The idea looks interesting.
(+) The experimental results look promising.

Cons)
(-) Many typos when denoting figures and tables. See the minor comments below.
(-) I believe the authors could organize the paper better. Tables and figures that are referred in a page are hard to find quickly. I recommend the authors refine the paper again for better readability.
(-) Some notations (such as RS-, RS-NAS, and so on) are so vague that hard to follow.
(-) I recommend the authors redraw all the figures for clarity. For example, each legend in Figure 2 is hard to take a look at.
(-) + the comments below.

Comments)
- When doing feature map resize in terms of the resolution, why Bilinear sampling was chosen? Could the authors provide a comparison with other sampling methods?
- In the related work section for dynamic neural networks, the authors claimed that "Most dynamic networks methods sacrifice accuracy in exchange of adaption in inference", but it seems to be quite overclaimed. As shown in the paper [1],  one can find that the author presented they could improve both accuracy and efficiency.
- How did you find the architecture shown In Figure 3 in the Appendix? What is Xception? Please specify the details.
-  Designing the pre-defined spatial list of L looks critical, so the authors should describe L in the implementation details.
- One of the main problem I think is the training budget issue. According to algorithm 1, the inner loop of "for j=0,..,len(L)", the overall training time will clearly take L times longer than that of the training setting w/o resizable training. Thus, it does not seem to be fair comparison in terms of the training budget. Namely, it seems that the authors compared with the other data augmentation methods which spend much less training budgets.
- Hard to grasp Section 3.5 of Adaptive resizable neural network. ResizeLearner looks being attached at the last stage of the original network after the original network is trained, but there is no further information about what ResizeLearner learns and how ResizeLearner selects the optimal sub-network.

[1] Universally Slimmable Networks and Improved Training Techniques, https://arxiv.org/pdf/1903.05134.pdf.

Minor comments)
- Wrong section and figure references:
  - 'It also mitigates the co-adaptation issue which we will discuss in Section 3.3'(indeed it is Section 3.4), 'The network architecture along with feature map resolution and channels number are shown in Figure 4' (it should be Figure 3).
  - - Figure 3(d) referred to in section 4.4  would be Figure 2(d) indeed.

About rating)
The authors provided a novel technique about the resizable approach and the experimental results look promising. However, the paper needs to be revised and looks like it does not ready to be published now. If the authors could revise the paper and concern my comments well, I would increase my rating.

**Experience Assessment:**

I have published in this field for several years.

**Review Assessment: Checking Correctness Of Derivations And Theory:**

I assessed the sensibility of the derivations and theory.

**Review Assessment: Checking Correctness Of Experiments:**

I assessed the sensibility of the experiments.

**Review Assessment: Thoroughness In Paper Reading:**

I read the paper thoroughly.

---

> ### Author Response · Authors · 2019-11-15
> **Response to Reviewer #1**
>
> Thanks for your detailed and thoughtful reviews! We have addressed all the questions below:
>
> We improved the writing and fixed the typos you have mentioned. We reorganized the position of tables and figures in our paper. We added subsection "notation" at the beginning of the experiment section to clarify the different words we used in our experiments. We enlarged the Figure 2 for readability.
>
> Q1: why Bilinear sampling was chosen? Could the authors provide a comparison with other sampling methods?
> A1: Bilinear sampling is chosen because it can downsample the feature map to any resolution.
> At this time, we are unable to finish the comparison of bilinear sampling with other sampling methods due to limited time constraints, but we will finish the experiments and add the ablation study in the final revision.
>
> Q2: It is overclaimed that "Most dynamic networks methods sacrifice accuracy in exchange of adaption in inference" in the related work section for dynamic neural networks.
> A2: You’re correct. We revised this section and included the paper [1] in the reference you have mentioned.
>
> Q3: How did you find the architecture shown in Figure 3 in the Appendix? What is Xception? Please specify the details.
> A3: The Xception is proposed by the paper [2], and the search space we use is described in [3]. We used the search method in paper [4]. We included this reference in our new revision. We did not discuss the details about how we find architecture because it is not the focus of our paper.
>
> Q4: Regarding the choice of pre-defined spatial list of L.
> A4: We added a subsection in method (Section 3) to define L. We included the choice of L we used in the experiments section. We added a new section in Appendix A to discuss the choice of pre-defined spatial list L.
>
> Q5: Unfair comparison with other data augmentation methods in terms of training budgets.
> A5: We first note that, for all resizable augmentation methods, we use the pre-defined spatial list of [0.45, 0.64, 0.85, 1.0], thus the len(L) = 4. Moreover, our training epochs is the same as the baseline, e.g., 100 epochs for ResNet-50 with batch size 512.
>
> Regarding the training budget, our resizable augmentation spends less or the same training budget compare to other data augmentation methods. Notably, data augmentation methods normally spend longer training time to improve generalization. For comparison, the baseline of ResNet-50 is trained by 100 epochs with batch size 512, AutoAugment [5] spend 21.6x longer training time (270 epochs with batch size 4096) and MixUp [6] spend 4x longer training time (200 epochs with batch size 1024) to train ResNet-50 compared to baseline, according to their paper. Resizable augmentation spend approximately 4x longer training time (100 epochs with batch size 512, but len(L) = 4). Thus the comparison with the other data augmentation methods in terms of training budget is fair.
>
> Q6: what ResizeLearner learns and how ResizeLearner selects the optimal sub-network?
> A6: We understand that the description in Section 3.5 is ambiguous. Thus, we made a revision on Section 3.5 (Section 4.3 in the new revision). To answer your question, in short, ResizeLearner learns the optimal scale configuration for each input image. The input of ResizeLearner is the feature map of an input image (process by the already trained ResizableNet), and the output is a scale configuration for this input. The ResizableNet can make predictions on this input use the scale configuration provided by ResizaLearner, thus make the inference data-dependent.
>
>
> [1] Yu, Jiahui, et al. Universally Slimmable Networks and Improved Training Techniques, In ICCV 2019.
> [2] Chollet, François Xception: Deep Learning with Depthwise Separable Convolutions, In CVPR 2017.
> [3] Guo, Zichao, et al. Single Path One-Shot Neural Architecture Search with Uniform Sampling, https://arxiv.org/abs/1904.00420.pdf
> [4] VAENAS: Sampling Matters in Neural Architecture Search, https://openreview.net/forum?id=S1xKYJSYwS
> [5] Cubuk, Ekin D., et al. AutoAugment: Learning Augmentation Policies from Data. In CVPR 2019
> [6] Zhang, Hongyi, et al. mixup: BEYOND EMPIRICAL RISK MINIMIZATION. In ICLR 2018

---

### Public Comment · ~Jason_Kuen1 · 2019-12-21
**Concerns about the novelty of this work**

I have great concerns about the novelty of this work and do not quite agree with the opinions of PCs and reviewers that this work is novel.

The essential idea of this paper is very similar to that of our "Stochastic Downsampling for Cost-Adjustable Inference and Improved Regularization in Convolutional Networks" paper published at CVPR 2018. The methods proposed by these two works essentially focus on training multiple sub-networks sharing the weights of a single full network based on different feature map resizing/downsampling configurations. The main objective of these two works is to enable test-time evaluation of different sub-networks (each has a different accuracy-efficiency operating point) obtained from a single model without re-training or fine-tuning. Both also share the identical idea of having separate Batch Normalization statistics for the different sub-networks -- known as Scale-Aware Batch Normalization in this work and Instance-Specific Batch Normalization in Stochastic Downsampling. Despite the strong similarities, this work does not mention or cite our Stochastic Downsampling paper at all.

The major difference between the two works is the "search" space for the feature map resizing/downsampling. Resizable Neural Networks resizes the feature map for each stage by randomly sampling a size (scaling ratio) from a predefined range of sizes. Whereas, Stochastic Downsampling samples a random layer index and a downsampling ratio for resizing the selected layer's feature map. I would say the fair sampling method introduced by this work is quite novel in the context of neural networks with multiple accuracy-efficiency operating points. But the accuracy gains shown in Table 10 (page 18) are apparently neither consistent nor significant.

I hope the authors will take the Stochastic Downsampling paper into good consideration and include it as a related work in the future version of this work.

---

### Decision · Program_Chairs · 2019-12-19

**Decision:**

Reject

**Comment:**

This paper offers likely novel schemes for image resizing.  The performance improvement is clear.  Unfortunately two reviewers find substantial clarity issues in the manuscript after revision, and the AC concurs that this is still an issue.  The paper is borderline but given the number of higher ranked papers in the pool is unable to be accepted unfortunately.